# Diurnal temperature range as a key predictor of plants' elevation ranges globally

Arnaud Gallou [1]✉, Alistair S. Jump [2], Joshua S. Lynn [3,4], Richard Field [5], Severin D. H. Irl [6], Manuel J. Steinbauer [1,7], Carl Beierkuhnlein [8,9], Jan-Chang Chen[10], Chang-Hung Chou[11], Andreas Hemp [12], Yohannes Kidane [8], Christian König [13,14], Holger Kreft [13,14], Alireza Naqinezhad[15], Arkadiusz Nowak[16,17], Jan-Niklas Nuppenau [18], Panayiotis Trigas [19], Jonathan P. Price[20], Carl A. Roland [21], Andreas H. Schweiger [22], Patrick Weigelt [13,14,23], Suzette G. A. Flantua [3] & John-Arvid Grytnes [1]

A prominent hypothesis in ecology is that larger species ranges are found in more variable climates because species develop broader environmental tolerances, predicting a positive range size-temperature variability relationship. However, this overlooks the extreme temperatures that variable climates impose on species, with upper or lower thermal limits more likely to be exceeded. Accordingly, we propose the 'temperature range squeeze' hypothesis, predicting a negative range size-temperature variability relationship. We test these contrasting predictions by relating 88,000 elevation range sizes of vascular plants in 44 mountains to short- and long-term temperature variation. Consistent with our hypothesis, we find that species' range size is negatively correlated with diurnal temperature range. Accurate predictions of short-term temperature variation will become increasingly important for extinction risk assessment in the future.

There is remarkable variation in species' range sizes, from endemic species confined to single mountaintops to cosmopolitan species occurring in most habitats around the world[1]. Although variation in species distributions has been highlighted since the earliest days of ecology and biogeography[2], the question of how range size dynamics relate to climate and geography has remained controversial[3–7]. Progress on this question is essential for better understanding ecological and evolutionary processes, broad-scale diversity patterns[8,9] and species' extinction risks in the context of climate change[10,11].

Temporal thermal variation is an extrinsic ecological factor that has often been used to explain variance in species' range sizes[5,12–16]. In his seminal contribution "Why mountain passes are higher in the tropics", Janzen[17] suggested that species experiencing large temperature fluctuations evolve broader thermal tolerances than species inhabiting ecosystems with relatively constant temperatures (Fig. 1a). Stevens[8,18] used this argument to explain observations that species' geographical ranges increase towards high latitudes or elevations (known as

Rapoport Rule)[19]. He hypothesized that the broader thermal tolerance of species inhabiting thermally variable environments allows these species to survive in a broader range of latitudes and elevations (Fig. 1b; hereafter "Stevens' hypothesis").

While a direct link between species' thermal tolerances, temperature variability and occupied geographic ranges is appealing, Stevens' hypothesis misses a key point. Stevens' hypothesis neglects lethal temperatures as a limit for any given species' range size and that temperature variability differs between locations. For example, a location with mean temperature of +5 °C on a mountain with high thermal variability may experience temperatures from -5 °C to +15 °C during the year or day (Fig. 1c). At that location, a species with a lethal lower temperature limit of 0 °C and upper limit of 20 °C will thus experience lethally cold temperatures frequently. Lower down, a similarly variable location with mean +20 °C will vary from +10 °C to +30 °C annually or daily. There, that same species will experience lethally high temperatures. The range of elevations in which the

species does not experience lethal temperatures in this thermally variable mountain is smaller than in an equivalent, more thermally constant mountain (Fig. 1c). Thus, greater temperature variability is a mechanism that will tend to reduce species' elevation ranges. This counteracts the expectation of Stevens' hypothesis (Fig. 1b) that larger-ranged species are expected to be found in climatically more variable mountain systems. It leads us to propose that the opposite pattern may instead be found (Fig. 1d). We call this the "temperature range squeeze" hypothesis. The squeeze of species' range sizes might not necessarily be symmetrical (as Fig. 1c suggests) or determined only by temperatures. Temperature variation could, for example, have a stronger influence toward the colder end (higher elevations) than towards the warmer end (lower elevations) of species' ranges, which may be codetermined by other factors such as competition. The underlying mechanisms behind the two hypotheses also highlight a key difference between the evolutionary-focused Stevens' hypothesis and the temperature range squeeze hypothesis that assumes no dependencies between thermal tolerance and temperature variation but where range sizes are determined by abiotic filtering. The pattern predicted by the temperature range squeeze hypothesis may be dampened by the larger thermal tolerances of species in more variable environments proposed by Janzen[17] or by avoidance strategies such as dormancy, but would not be reversed unless the increase in tolerance over-compensates for the range squeeze.

Here we assess the contrasting predictions of how range size scales with temperature variability (herein we use this term for temporal thermal variability) using a global dataset of more than 88,000 elevation range size estimates for vascular plants in 44 mountains (29 continental and 15 island mountains; Supplementary Fig. 1). Each range size estimate is the difference between the maximum and minimum species' elevations in a given mountain (see Methods and Supplementary Table 1 for details). 'Mountains' are defined as mountainous areas, regions or countries, mountain ranges or volcanoes with an elevation span ≥1500 m. We used diurnal and seasonal temperature variations (mean diurnal range of temperatures averaged over one year and standard deviation of the monthly mean temperatures, respectively) as the main predictors of species' range sizes, as originally suggested by Stevens[18]. Very few studies have tested Stevens' predictions at a temporal scale other than among-season variations.

Using multiple temporal scales to investigate the relationship between temperature variability and species' range sizes is judicious because the frequency at which species experience high variation in temperature are likely to lead to different responses among species. For instance, many plant species inhabiting temperate climates are able to escape cold temperatures via dormancy. The effective temperature range experienced by such species should then be smaller than that of the full range of temperature actually occurring during the year. On the contrary, species growing in habitats where temperatures vary greatly during the day have no choice but to cope with the full range of temperature experienced in their vicinity. Plant responses to temperature variation could then be expected to be stronger as the temporal scale shortens. Because most plants are perennial and experience similar diurnal and seasonal temperatures during their lifespan (within-generation variation)[20], we also tested the influence of temperature variation in the last 2000 years (from 0 to 1980 AD; $\Delta MAT_{0-1980}$) on species' range sizes (among-generation variation)[20]. Most previous studies tested the relationship of plant species' elevation ranges with thermal variation along single mountain gradients, which makes the decoupling of climatic and geographic factors difficult. These limitations are best addressed with global-scale analyses and multiple elevation gradients: our core analysis. In addition, for direct comparability with Stevens' original hypothesis and previous studies, we tested the response of species' elevation ranges along each elevation gradient.

## Results and Discussion

We asked how diurnal temperature range, temperature seasonality, and $\Delta MAT_{0-1980}$ predict the elevation ranges of vascular plants averaged within mountains across elevation gradients (global-scale analyses). Diverging conclusions in previous studies that investigated the relationship between temperature variability and species' range sizes have been attributed to methodological issues, such as sampling effort[21], geometric constraints[4,7] and analytical methods[22,23]. To minimize the influence of these factors, we standardized the length of each elevation gradient to 2500 m and discarded species found exclusively in the upper and lower 250 m (exclusion zones; see Methods and Supplementary materials for details and justifications). We ran parallel analyses using standardized gradient lengths of 1500 and 2000 m and exclusion zones of 0 and 500 m, which did not influence the main

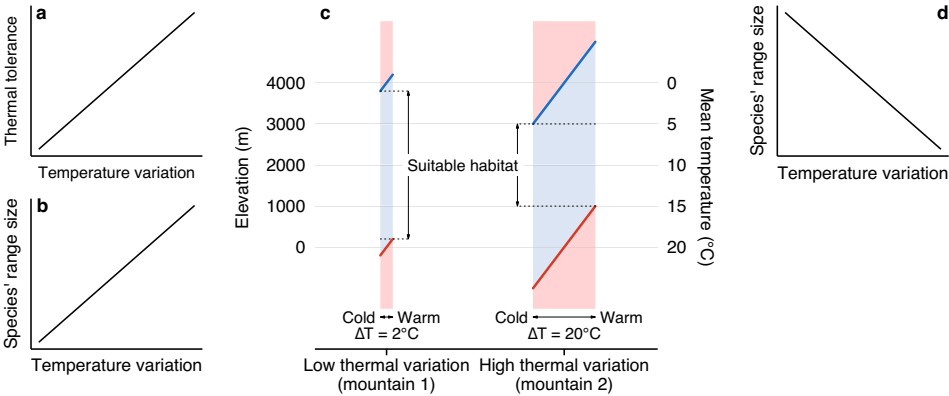

**Fig. 1 | Illustration of Stevens' hypothesis and the temperature range squeeze hypothesis.** Based on the assumption that species have larger thermal tolerances in climatically variable habitats than in less climatically variable ones (**a**), Stevens predicted a positive relationship between species' range sizes and temperature variability (**b**). Panel **c** represents the spatio-temporal temperature variation in mountains with different levels of temperature fluctuation as explained in the temperature range squeeze hypothesis. For simplicity, we used a constant lapse rate of 0.5°C. Shaded areas represent tolerable (blue) and lethal (red) temperatures for a species' thermal tolerance spanning from 0 °C to 20 °C. The widths of the shaded areas represent temperature variability over time (ΔT), while the spatial variation in temperatures is described by the vertical axes. Thick blue and red lines indicate the 0 °C and 20 °C isotherms, respectively. The suitable habitat, which corresponds to species' maximum theoretical elevation range in the absence of avoidance strategies such as dormancy, is defined by elevations in which the species does not encounter a limiting temperature (i.e. below 0 °C or above 20 °C) at any time of the given time scale. The elevation span of the suitable habitat is expected to shrink as thermal variability increases (**c**, compare mountains 1 and 2). Thus, a negative relationship between species' range sizes and thermal variation is expected (**d**).

results (see Supplementary materials for results using different data constraints).

Species' elevation range declined with all measures of temperature variability used in the study (Fig. 2), contradicting Stevens' hypothesis and supporting the temperature range squeeze hypothesis. In addition, models using temperature variability alone performed better than models including interactions with annual precipitation and mean annual temperature (Supplementary Table 2), suggesting that additional climatic variables did not increase the predictive ability of the model. A higher probability for species to encounter limiting temperatures sooner as they move away from their optimum habitat likely explains the reduced elevation range sizes observed in thermally variable mountains. The influence of extreme temperatures could be accentuated by small population sizes and gene swamping towards species' range ends which may limit the capacity of such populations to adapt to thermal extremes[24,25].

Among thermal predictors, diurnal temperature range was the most predictive (lowest WAIC and LOO; Supplementary Table 2) and had the strongest and least uncertain relationship to species' range size ($P(\beta < 0) = 1$, $R^2$: 0.43), followed by temperature seasonality ($P(\beta < 0) \approx 0.96$, $R^2$: 0.12) and $\Delta MAT_{0-1980}$ ($P(\beta < 0) \approx 0.97$, $R^2$: 0.12; Fig. 2, Supplementary Fig. 7 & Supplementary Table 3). The strong relationship of species' range size to diurnal temperature range suggests that short-term temperature variation is more important in determining species range sizes than previously thought. Longer-term variables such as temperature seasonality might be less relevant than expected due to the ability of plant species inhabiting seasonal climates to avoid long periods of unfavorable temperatures via dormancy. As a consequence, the relative range of temperatures effectively experienced by extratropical species may be lower than the estimated seasonal variability. The narrowing of sessile species' elevation ranges may be exacerbated in mountains with high diurnal temperature variation because they cannot escape extreme temperatures over such short time scales. A decline in species' elevation ranges in response to high diurnal temperature range has also been reported in terrestrial vertebrates[22], suggesting a general pattern. Chan et al.[22] used a simulation by Gilchrist[20] to explain the negative relationship via evolutionary selection by diurnal temperature fluctuation for narrow thermal tolerance. Gilchrist's simulation assumes the survival of all individuals under any circumstances and variability in range size is merely linked to how reproductive success is related to performance breadth[20]. Predictions are thus only realistic if species survive unsuitable temperatures (e.g. via dormant stages). If this assumption is not met, species with small elevation ranges need broad thermal tolerance in order to survive the extremes of short-term temperature fluctuation (Supplementary Fig. 10). Gilchrist's hypothesis predicts that performance breadth increases when high among- and low within-generation

variation occur at the same time, while the combination of low within- and among-generation variation would lead to smaller performance breadth and consequently to smaller range sizes. Exploring our data further to assess the influence of these interactions did not show support for Gilchrist's hypothesis (Supplementary Figs. 11–12).

Island diversity differs from mainlands as a result of isolation-driven immigration, extinction and speciation processes[26], which could influence species' range sizes differently. Comparing species' range sizes in island and continental mountains revealed broader mean species' range sizes on islands (Fig. 2). However, diurnal temperature range was the only tested variable to account for variation in species' range sizes between island and continental mountains. Island mountains were characterized by both lower diurnal temperature range and broader mean elevation range sizes than continental mountains (Fig. 3). Lower diurnal temperature variation in island mountains likely results from a buffering effect of the ocean on air temperature that would make the climate less temporally variable[27,28]. The strength and continuity in the diurnal temperature range-species' range size relationship, with a negative trend being visible in both island and continental mountains (Fig. 3a), suggest that short-term temperature variation plays a major role in driving the observed range-size pattern that cannot be imputed to island-specific factors (e.g. lower competition). The temperature range squeeze hypothesis thus offers a unifying explanation for elevation range sizes on continents and islands at a global scale.

We also assessed the relationship between temperature variation and species' elevation range within each individual mountain (local-scale analyses) using diurnal temperature range and temperature seasonality. We did not test the influence $\Delta MAT_{0-1980}$ on species' elevation range for the local-scale analyses because of the low spatial resolution of the past climate data. Because each mountain gradient was tested independently, we did not standardize elevation gradients. For the same reasons mentioned in the global-scale analyses, we used exclusion zones of 250 m and we ran parallel analyses using exclusion zones of 0 and 500 m (see Supplementary materials for results using different data constraints).

Results from the local-scale analyses were less conclusive than the analyses at a global scale, with about as many mountains featuring negative as the positive influence of diurnal temperature range (45% positive vs 55% negative, including 34% positive vs 39% negative with low uncertainties) or temperature seasonality (52% positive vs 48% negative, including 41% positive vs 34% negative with low uncertainties) on species' elevation ranges (Fig. 4, Supplementary Table 4). The lack of a clear trend in the direction of the estimates for all elevation gradients suggests that, at a local scale, temperature variability might not be the dominant driver of species' elevation ranges, or has different influences depending on plant community composition. Our

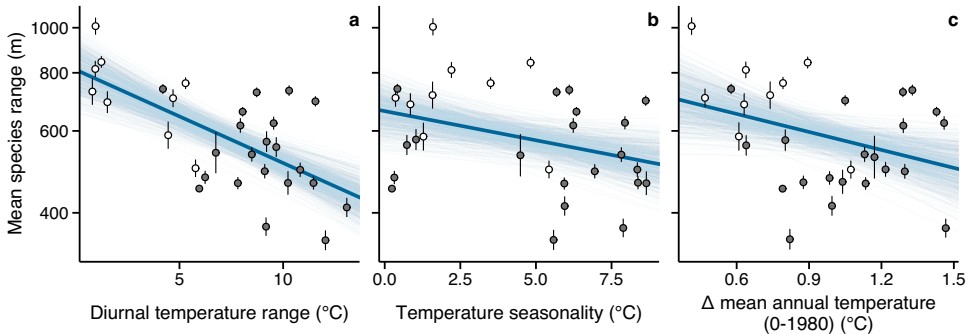

**Fig. 2 | Relationships between thermal variability and mean species' elevation ranges.** Diurnal temperature range (**a**), temperature seasonality (**b**) and the variation of mean annual temperature from 0 to 1980 AD (**c**). Points represent the estimated mean elevation ranges with their respective standard error in each of the 30 standardized elevation gradients with length ≥ 2500 m. Thick blue lines are the posterior mean calculated from 600 random draws sampled from the 95% credible interval (thin blue lines). Colored dots indicate island (white) and continental (dark gray) mountains. Source data are provided as a Source Data file.

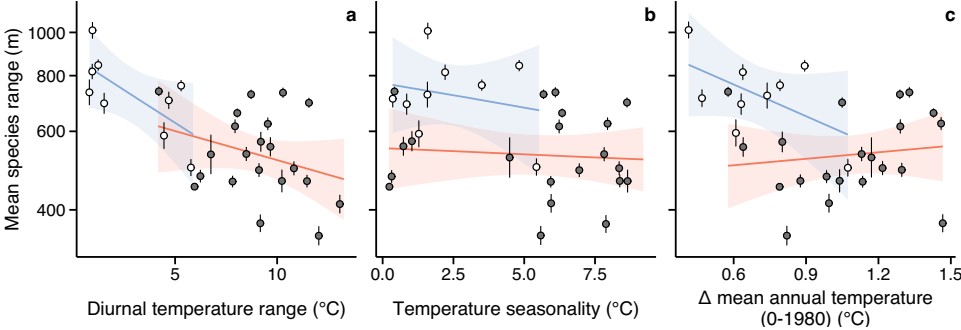

**Fig. 3 | Relationships between thermal variability and mean species' elevation ranges in island and continental mountains.** Diurnal temperature range (**a**), temperature seasonality (**b**) and the variation of mean annual temperature from 0 to 1980 AD (**c**). Points represent the estimated mean elevation ranges with their respective standard error in each of the 30 standardized elevation gradients with length ≥ 2500 m. Colors indicate island (white dots, blue regressions) and continental (dark gray dots, orange regressions) mountains. Source data are provided as a Source Data file.

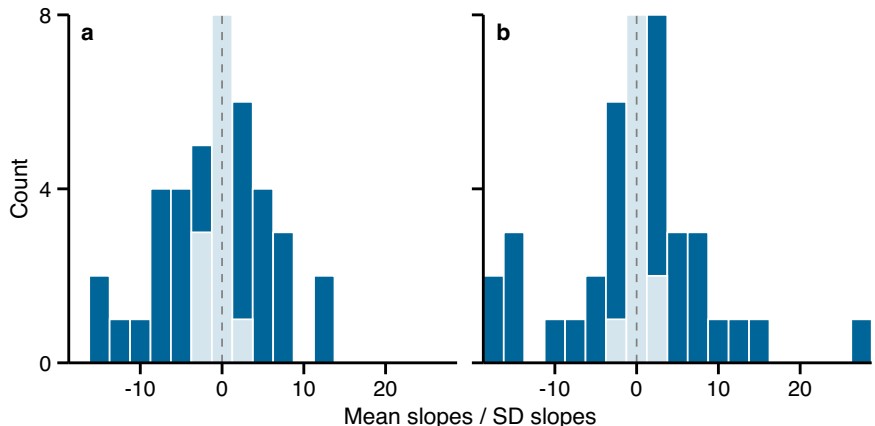

**Fig. 4 | Results from the local-scale analyses.** Responses of species' elevation ranges to diurnal temperature range (**a**) and temperature seasonality (**b**) within each of the 44 mountains used in the study. Histograms are of mean slope terms divided by their standard deviation from the 95% credible interval for the local-scale analyses. Colors indicate whether the 95% credible interval includes 0 (light blue) or not (dark blue). Dashed lines indicate 0. Source data are provided as a Source Data file.

results thus potentially explain ambiguous conclusions from previous studies on the role of thermal variation in shaping species' elevation ranges along individual mountain gradients. The small variation in our data on diurnal and seasonal temperature range within single mountain gradients compared to the variations across mountains could explain the weak influence of thermal variability in driving species' elevation ranges locally (Fig. 4, Supplementary Fig. 13). This could be exacerbated in the local-scale analyses by the resolution of the climate data that might hide spatial heterogeneity in temperature variation along single elevation gradients[29]. Additionally, species' ranges are likely to be co-determined by multiple factors that are mountain-dependent, such as human disturbances, soil type, water availability, dispersal abilities, ecological history, microgeography or interspecific interactions[6,21,24,25,30–34].

The ratio of positive and negative relationships between species' range sizes and temperature variation remained essentially the same, independent of the size of exclusion zones (Supplementary Fig. 14, Supplementary Table 4). However, the number of slopes with low uncertainties decreased drastically with increasing exclusion zone sizes (Supplementary Fig. 14, Supplementary Table 4), suggesting that species found exclusively at the domain boundaries strongly influence the strength of the pattern within mountains and that the narrowing of species' range sizes at domain boundaries greatly influence the probability of finding statistically significant relationships. Species exclusively inhabiting lower and higher elevations had less influence in the

global-scale analyses, where results remain consistent, independent of the size of the exclusion zone.

In summary, our findings solidify the temperature range squeeze hypothesis (Fig. 1c–d) that predicts a decline in species' range sizes in climatically variable habitats, thereby contradicting Stevens' hypothesis[18] (Fig. 1b). Our study reveals that the variation in diurnal temperatures might play a determining role in shaping the elevation ranges of vascular plants on a global scale and contributes to the differences in range sizes observed between continents and islands. Local (within-mountain) variation in species' elevation ranges, however, is largely decoupled from temperature variability and could be the result of local, interacting variables, such as species interactions, land use, microclimatic variations and soil type, among others. The detected global signal of the role of variation in diurnal temperature in shaping elevation ranges urges the need to reconsider past theories on our understanding of the driver of plant species distributions. Especially for the field of conservation biology in the face of global change, these novel insights are relevant to take into account. Our findings call into question the prevailing understanding that global changes will especially imperil tropical species with restricted ranges[35]. We suggest that extinction risk may be higher in continental mountains where species are more likely to have smaller elevation ranges due to higher thermal variability than species inhabiting mountain systems exposed to oceanic climates. Short-term temperature variation may also become increasingly important in driving local extinction risks but a large

variation of local variables will need to be considered. Considering our findings, the temperature range squeeze hypothesis holds true on a global scale, shedding light on the determining influence of diurnal temperature variation on species' elevation ranges and emphasizing the need for adaptive conservation measures in the context of a rapidly changing climate.

## Methods
### Plant data
We compiled a global dataset of vascular plant species with elevation ranges from published and unpublished data (online repositories, checklists, floras as well as private and museum collections), including more than 96,000 species' elevation ranges in 56 mountains. We restricted our search to elevation gradient length ≥1500 m and mountain regions spanning less than 1000 km along a North-South and East-West axis, for which elevation range size data were given explicitly for the relevant mountain areas at a precision ≤100 m. The elevation ranges were directly provided for every species as a minimum and maximum elevation in all but Mt. Kilimanjaro, Mt. Etna and Switzerland datasets, for which elevation ranges were computed from occurrence records. Species range sizes in Switzerland were computed from GBIF observations between 1980 and 2021 (see Supplementary Table 1 and references for details) that we curated using the CoordinateCleaner package version 2.0.20[36] to remove species around the GBIF headquarters, institutions, capital, centroids, outliers, records that fall into the ocean, zeros and absolute latitude and longitude. We defined the elevation range of species as the difference between the maximum and minimum reported observation in each elevation gradient.

We implemented several steps to homogenize the data and reduce biases related to dissimilarities in sampling intensity among elevation gradients. First, we discarded mountains with discontinuous sampling, defined as one or more gaps >500 m between two sampled sections within the same elevation gradient. Second, we calculated the percentage of singletons−i.e. species with only one observation and a consequent elevation range of 0 m−in each dataset to assess the sampling effort for every elevation gradient. A high proportion of singletons might reflect a poor sampling effort and would result in highly underestimated species' range sizes. We retained mountains with ≤ 25% singletons, which provided a compromise between the number of elevation gradients and the variance in sampling intensity in the data. Third, we removed a total of 38 observations with obviously incorrect elevation values, e.g. species with a minimum elevation greater than the maximum elevation or with an elevation higher than 6500 m, corresponding to the highest elevation recorded for vascular plants[37]. These erroneous data were present in the original source and probably resulted from typographic errors, such as additional digits in elevation values. Finally, we standardized taxon names to the species level using GBIF's species name matching tool[38]. Taxa that could not be identified to the species level were discarded. The final dataset consisted of more than 88,000 range size data and 44 elevation gradients.

### Climate data
We used global climate data from CHELSA[39,40] with a spatial resolution of 30″, covering the period between 1979 and 2013, to investigate the response of species' elevation range sizes to diurnal temperature range (DTR; bio2) and temperature seasonality (TS; bio4), defined as the mean diurnal range of temperatures averaged over one year and standard deviation of the monthly mean temperatures, respectively. Additionally, we used the mean annual temperature (MAT; bio1) and annual precipitation (AP; bio12) to explore the interacting influence of MAT and AP with temperature variability on range sizes. To compute mean bioclimate values within 100 m elevation bands, we first generated elevation bands by reclassifying SRTM rasters (1 arc-sec resolution) downloaded from the U.S. Geological Survey[41]. SRTM rasters

covered the entire region of interest, which could be an entire mountain, mountain range or administrative unit (e.g. state, province or country). We then resampled the climate rasters to the resolution of the SRTM rasters so that each pixel between the climate and SRTM rasters matches one another. Finally, we averaged values from the climate rasters for each elevation band in every mountain using the 'zonal' function from the R package terra (version 1.6.7)[42].

To examine the influence of temperature variability in the last 2000 years ($\Delta MAT_{0-1980}$) on species' ranges, we generated a time series of global annual mean temperature between 0 to 1980 (AD) using the PaleoView software version 1.5.1[43], which provides reconstructions of past climate at high temporal resolutions. The time series was generated using 30-year intervals taken in 30 year steps and the bias correction turned off. We defined $\Delta MAT_{0-1980}$ as the difference between the highest and lowest mean annual temperature values in the time series in each mountain gradient. Because the spatial resolution of the simulated past climate data ($2.5 \times 2.5°$) was too low to conduct analyses along elevation, we extracted $\Delta MAT_{0-1980}$ values independently of the elevation (i.e. one $\Delta MAT_{0-1980}$ value per elevation gradient).

### Statistical analyses
We investigated the effect of temperature variability on species' elevation ranges with two Bayesian models. First, we asked how temperature variability predicts mean elevation ranges of vascular plants across mountains (global-scale analyses), using diurnal, seasonal and temperature variation from 0 and 1980 (AD). Because most studies investigating the relationship between species' elevation ranges and temperature variation are done within single mountain gradients, we also assessed how DTR and TS predict species' elevation ranges at a local scale along each elevation gradient. To have the data meet normal distribution assumptions, we set a range size of 10 m−corresponding to the smallest species' elevation range in our dataset−to all singletons and applied a natural log transformation of range sizes prior to the analyses.

### Global-scale analyses
To test the overall influence of temperature variability on species' elevation ranges, we fit models predicting the response of species' ranges to DTR, TS and $\Delta MAT_{0-1980}$. We averaged species' range sizes and bioclimate values from every elevation band within each mountain (i.e. every mountain is described by one species' range size and one bioclimate value). The range of thermal variation captured within each single mountain was relatively small in comparison to the range of thermal variation across mountains (Fig. 2 and Supplementary Fig. 12). Thus we expect the largest variation in species' range sizes to occur in the global-scale analyses rather than the local-scale analyses. Additionally, we ran models to test the AP:DTR, MAT:DTR and $MAP_{0-1980}:\Delta MAT_{0-1980}$ interactions on mean species' range sizes. We did not test the AP:TS or MAT:TS interactions because of their strong correlation (-0.67 and -0.65, respectively).

Mountain gradients in the dataset varied between 1500 and 6430 m. Such disparities in the length of elevation gradients can directly influence species' range sizes (Supplementary Fig. 8). For instance, a short gradient in elevation is more likely to display narrow species' ranges, simply because species are more strongly constrained by the upper and lower limits of the elevation gradient. Similarly, we would expect to observe narrower species' ranges toward the ends of the elevation gradients because the physical barriers created by the domain limits will stop species from expanding beyond the domain's boundaries (or will truncate species' range if the sampled gradients does not cover the full elevation gradients). As a result, species thriving close to the edges of the elevation gradients are likely to be truncated and display a fraction of their potential range[44,45].

To reduce biases related to the length and limits of elevation gradients, we standardized the length of each elevation gradient by

retaining a set of elevation ranges at the top of each mountain gradient and deleting the rest. For example, for a standardized elevation gradient length of 2000 m, an original elevation gradient running from 0 to 5100 m a.s.l. would be converted into an elevation gradient going from 3100 to 5100 m a.s.l. This approach ensures comparability between elevation gradients. We truncated species' elevation ranges crossing the lower end of the standardized elevation gradients, keeping the parts of the species' ranges within the gradient's boundaries only. Species whose distribution range was entirely outside the standardized mountain gradient limits were discarded from the analyses. To minimize the influence of truncated species at the elevation gradient edges on the analyses, we excluded species found exclusively near the top and bottom of each elevation gradient using an exclusion zone.

To assess the sensitivity of the model outputs to these methodological choices, we ran each model using standardized gradient lengths of 1500, 2000 and 2500 m, and exclusion zones of 0, 250 and 500 m. We also repeated analyses with elevation gradients standardized from the bottom of the mountain gradient (rather than the top). These parallel analyses produced qualitatively similar results for gradient lengths varying from 1500 and 2500 m (Supplementary Figs. 2 and 5). Longer standardized elevation gradients would discard too many mountains, thereby limiting the reliability of the model estimates. For the results presented in the main text, we used standardized gradients of 2500 m ($n = 30$) and exclusion zones of 250 m as these values provided a good compromise between the number of elevation gradients to fit the model and species range data to estimate the average elevation range in each vertical gradient.

The Bayesian hierarchical models[46] first estimated the mean elevation range of species within each mountain, and then fit linear models of mean range sizes by mean climate of the range with:

$$RS \sim N(\mu_{RS}, \sigma_{RS}^2) \tag{1}$$

$$\mu_{RS} = \alpha_{site}[site] \tag{2}$$

$$\sigma_{RS}^2 \sim \Gamma(10^{-3}, 10^{-3}) \tag{3}$$

$$\alpha_{site} \sim N(\mu_{site}, \sigma_{site}^2) \tag{4}$$

$$\mu_{site} = \alpha_V + \beta_V * V_{site} \tag{5}$$

$$\alpha_V \sim N(0, 10^{-6}) \tag{6}$$

$$\beta_V \sim N(0, 10^{-6}) \tag{7}$$

$$\sigma_{site}^2 \sim \Gamma(10^{-3}, 10^{-3}) \tag{8}$$

Where species range size ($RS$) is a normally distributed random variable (with mean, $\mu$, and variance, $\sigma^2$; eq. (1)) summarized with mountain (site) specific means and variances with $\alpha_{site}$ (eq. (2)). Eq. (3) is a gamma prior for the variance in species range sizes in eq. (1). The uncertainty in $\alpha_{site}$ is propagated through to eq. (4) and eq. (5), which models the relationship between mean range size of species within a mountain ($\mu_{site}$) and a given climate variable $V$ (representing either DTR, TS, $\Delta$MAT$_{0-1980}$ or the interaction between temperature variability and TS, AP or land type) with intercept ($\alpha_V$) and slope ($\beta_V$) terms. Eq. (6) and eq. (7) are normal, flat priors for slope and intercept terms in eq. (5) and eq. (8) is a flat gamma prior for the variance in mean mountain range sizes ($\sigma_{site}^2$). We compared within-sample predictive ability of the

different climate variables using the Watanabe Akaike information criterion (WAIC)[46], and leave-one-out cross validation (LOO) from the loo package version 2.5.1[47] to determine which climate variables best predict species' elevation range sizes. Both criteria assess within-sample predictive error of the models. We evaluated strength and support for a given parameter or relationship through inspection of posterior probability distributions with a 95% credible interval[48].

### Local-scale analyses
These analyses examine the responses of species' range sizes to DTR and TS along each mountain gradient. We did not test the influence of $\Delta$MAT$_{0-1980}$ due to the low resolution of the estimated past climate data. Similarly to the global-scale analyses, we excluded species found exclusively in the lower and upper 250 m of each elevation gradient to reduce the influence of truncated species at the gradient edges (outputs of these analyses with 0, 250 and 500 m exclusion zones are presented in Supplementary Fig. 13). However, we did not standardize elevation gradients for these analyses because each elevation gradient was tested independently of the others. The length of elevation gradients showed no significant influence on slope direction within each mountain (Supplementary Fig. 12). As a result, we used all 44 mountain locations with an elevation span ≥ 1500 m. Finally, we assigned DTR and TS values to species' midpoint[49]. We modeled the responses of elevation ranges within each mountain as:

$$RS \sim N(\mu_{RS}, \sigma^2) \tag{9}$$

$$\mu_{RS} = \alpha[site_i] + \beta[site_i] * V \tag{10}$$

$$\alpha[site_i] \sim N(0, 10^{-6}) \tag{11}$$

$$\beta[site_i] \sim N(0, 10^{-6}) \tag{12}$$

$$\sigma^2 \sim \Gamma(10^{-3}, 10^{-3}) \tag{13}$$

Where species range size ($RS$) is a normally distributed random variable (with mean, $\mu_{RS}$, and variance, $\sigma^2$; eq. (9)) predicted by mountain-specific intercepts $\alpha[site_i]$ and slopes $\beta[site_i]$ for a given explanatory variable $V$ (representing either DTR or TS) where $i$ is one of the 44 elevation gradients (eq. (10)). Eq. (11) and eq. (12) are normal, flat priors for slope and intercept terms in eq. (10). Eq. (13) is a flat gamma prior for the variance in range sizes ($\sigma^2$). We assessed strength and support for each given parameter with a 95% credible interval.

### Model diagnostics
We fit the models using Markov chain Monte Carlo (MCMC) with R2jags version 0.7.1[50] in R version 4.0.2[51]. We ran 3 chains with 50,000 iterations each and a burn-in of 20,000 until the effective sample size for each parameter reached 3,000. We considered good convergence of the MCMCs to be when the potential scale reduction factor $\hat{R}$ was ≤ 1.01. We assessed good mixing of the models through visual inspection of trace and autocorrelation plots. Finally, we evaluated model fit using posterior predictive checks.

### Reporting summary
Further information on research design is available in the Nature Portfolio Reporting Summary linked to this article.

## Data availability
Data used in this study are accessible at https://doi.org/10.17605/OSF.IO/D42JQ[52] or in Supplementary Data 1 and 2. Source data are provided with this paper.

## Code availability

Code to reproduce the results and figures is available on OSF at https://doi.org/10.17605/OSF.IO/D42JQ[52].

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

## Acknowledgements

We are grateful to Monika Kriechbaum and Empar Carrillo Ortuño for providing us with species' authorities for accurate taxa standardization of the Mustang and Espot y Boí data, respectively. We thank Kirsten O'Sullivan, Chi-Hua Chang and the Forest Management Laboratory at the National Pingtung University of Science and Technology for translating, processing and putting together the Taiwan data. We thank Cathy Jenks for putting together the Drakensberg, Jamaica, Jaya and Hengduan datasets as well as Marten Winter for putting together the Afghanistan dataset. We thank Richard J. Telford for providing advice on the R code and figures as well as Camila Pacheco-Riaño for advice on GIS analyses. S.G.A.F. acknowledges support from Trond Mohn Stiftelse (TMS) and University of Bergen for the startup grant 'TMS2022STG03', and from the European Research Council (ERC) under the European Union's Horizon 2020 research and innovation program (grant agreement no. 741413) to H.J.B. Birks.

## Author contributions

A.G., J.S.L., R.F., S.D.H.I., M.J.S., C.B., S.G.A.F. and J.-A.G. designed the study; J.-N.N. conducted the preliminary study; A.G., A.S.J. and J.-A.G. conceptualized the temperature range squeeze hypothesis; A.G., J.S.L., S.G.A.F. and J.-A.G. conceived the methodology; A.G., J.-C.C., C.-H.C., A.H., Y.K., C.K., H.K., A.Na., A.No., J.-N.N., P.T., J.P.P., C.A.R. and P.W. collected and curated the data; A.G. conducted the investigation, performed the analyses and produced the figures; S.G.A.F. and J.-A.G. supervised and administrated the project; A.G. wrote the manuscript with contributions from A.S.J., J.S.L., R.F., S.D.H.I., M.J.S., A.H., C.K., H.K., A.No., J.-N.N., A.H.S., P.W., S.G.A.F. and J.-A.G.

## Funding

## Competing interests

The authors declare no competing interests.

## Additional information

[1]Department of Biological Sciences, University of Bergen, PO Box 7803, 5020 Bergen, Norway. [2]Biological and Environmental Sciences, Faculty of Natural Sciences, University of Stirling, FK9 4LA Scotland, UK. [3]Department of Biological Sciences, University of Bergen and Bjerknes Centre for Climate Research, Bergen, Norway. [4]Department of Earth and Environmental Sciences, The University of Manchester, Manchester, UK. [5]School of Geography, University of Nottingham, Nottingham NG7 2RD, UK. [6]Biogeography and Biodiversity Lab, Institute of Physical Geography, Goethe-University Frankfurt, Altenhöferallee 1, 60438 Frankfurt, Germany. [7]Bayreuth Center of Ecology and Environmental Research & Department of Sport Science, University of Bayreuth, 95447 Bayreuth, Germany. [8]Chair of Biogeography, University of Bayreuth, 95440 Bayreuth, Germany. [9]Department of Botany, University of Granada, Granada, Spain. [10]Department of Forestry, National Pingtung University of Science and Technology, Pingtung, Taiwan. [11]Institute of Plant and Microbial Biology, Academia Sinica, Taipei, Taiwan. [12]Department of Plant Systematics, University of Bayreuth, 95440 Bayreuth, Germany. [13]Biodiversity, Macroecology & Biogeography, University of Göttingen, Büsgenweg 1, 37077 Göttingen, Germany. [14]Centre of Biodiversity and Sustainable Land Use, University of Göttingen, Büsgenweg 1, 37077 Göttingen, Germany. [15]Department of Plant Biology, Faculty of Basic Sciences, University of Mazandaran, P.O. Box: 47416-95447, Babolsar, Iran. [16]Institute of Biology, University of Opole, Oleska St., 45-052 Opole, Poland. [17]PAS Botanical Garden - Center for Biodiversity Conservation in Powsin, Prawdziwka St. 2, 02-952 Warszawa, Poland. [18]Department of Ecology, Environment and Plant Science, Stockholm University, 106 91 Stockholm, Sweden. [19]Department of Crop Science, School of Plant Sciences, Agricultural University of Athens, Iera Odos 75, 11855 Athens, Greece. [20]Department of Geography, University of Hawaii, Hilo, Hawaii, USA. [21]Denali National Park, 4175 Geist Road, Fairbanks, AK 99709, USA. [22]Institute of Landscape and Plant Ecology, Department of Plant Ecology, University of Hohenheim, Ottilie-Zeller-Weg 2, 70599 Stuttgart, Germany. [23]Campus-Institut Data Science, University of Göttingen, Göttingen, Germany. ✉e-mail: arangacas@gmail.com

