## [Peer Review File · Nature Communications]

Diurnal temperature range as a key predictor of plants' elevation ranges globallyREVIEWER COMMENTS

Reviewer #1 (Remarks to the Author):

This is very nice study, clearly presented and convincing. The findings indicate that, in contrast to previous ideas, elevational range sizes of species are smaller when there is higher temporal (especially diurnal) variation. I do not have substantial comments, only two major things which might be useful to consider.

First, trends of range sizes along gradients (e.g. latitudinal or altitudinal) are subjects to several geometrical effects which distort the patterns. Most importantly, domain boundaries may truncate potential ranges given by climatic tolerance, so that observed range size patterns differ from the patterns comprising potential ranges. In this respect, it may be useful to consider paper of Sizing et al. (2009; Ecology 90: 3575-3586) which explicitly treats these issues (albeit on a latitudinal gradient, but the principle is universal). I acknowledge the authors have been aware about these issues and tried to avoid the effect of range truncations by standardizing elevational ranges and not counting species whose ranges are located just next to the domain boundary. Still, it is not clear if these measures completely remove these effects. Since I do not have any better recipe than what has been done by the authors, I would like at least to see whether there is an effect of the length of elevational gradient on the range size, and/or map of the locations with the indication of the lengths of the gradients (before standardization, of course) and mean range size (in addition to the map which is already presented in Extended data). For instance, are tropical localities those with longer elevational gradients and at the same time shorter ranges? I still think that shorter elevational gradients may shrink species ranges, even if the gradients are standardized by using only the upper part of the gradient with the standard size.

Second, the authors seem to assume that the range sizes are determined by limits of temperature tolerance. But it seems to me that temperature may limit only the upper range edge, whereas the low-elevation edge is often determined by interspecific competition and interactions generally (including natural enemies, parasites etc.) – species often can potentially live in lower elevations, but they are outcompeted by better adapted species there. How this go together with the reported patterns? Also, as far as I understand, daily

variation has been estimated for the whole mountain(s) as the data unit. But daily temperature variation changes along elevational gradients, so that range limits in different elevations are driven by different factors. Again, at the moment I do not have an idea how to resolve these issues, but I just think there may be effects not considered by the authors, although I appreciate the way how the authors have dealt with some confounding effects.

Minor comments:

Fig. 2: Continental mountains seem to be marked by black, not gray circles.

Line 119: Should it sound '...maybe lower..' rather than '...may be less'?

Methods, lines 6-7: This repeats what has been already in the main text.

Reviewer #2 (Remarks to the Author):

This work uses a global dataset of plant distributions in mountains to examine the relationships between diurnal, seasonal, and longer-term temperature variability and plant elevational range size, and finds a strong relationship between elevational range size and diurnal temperature variation. The plant distributional data collated for this work was impressive, and the analyses were well-explained and statistically robust.

My major criticism of this work was the framing, which sets up the longstanding, but often criticized, theory for a positive variability-range size relationships (Janzen/Steven) against a newly presented Temperature Squeeze Hypothesis. Although this new hypothesis is well-explained with Figure 1, the comparison between the only these two hypotheses seemed limited in terms of the large number of potential drivers of plant range sizes (Sheth et al. 2020 New Phytologist). Secondly, I was struck by the lack of exploration of the different processes (evolutionary vs. ecological) that are largely invoked for driving these two hypothesized patterns, with the former focused on the evolved response of thermal tolerance and the latter focused on ecological limitation through abiotic filtering of species with set thermal tolerances. It is well recognized that range limits are likely driven jointly by

evolutionary and ecological dynamics (Sexton et al. 2009), and this possibility it noted by the authors implicitly in L77-79. The current framing of the two hypotheses doesn't make it clear enough that one hypothesis is more of an evolutionary argument while the other is more ecological where thermal niches are assumed to be a stationary species-level characteristic.

I don't think that this adjustment of the framing would reduce the novelty of this work or lack of support for the findings. Range limit theory in papers already cited in the work present many reasons (low population size, gene swamping etc.) why range edge populations may not necessarily be able to adapt to the thermal extremes experienced in those populations (even if considering niche limits not to be lethal), and therefore abiotic filtering may be a larger driver of range sizes in mountains.

Further, the finding of the relationship between diurnal temperature variability and elevation range size is an exciting one, and I think that the introduction and work overall could have a stronger focus on these multiple scales of variability explored, the predictions for relationships between variability and range size at these different scales, and well as the implication for the patterns found. The authors argue how this type of variability is difficult to compensate for both through avoidance (e.g., dormancy) and adaptation (as it may be less predictive than seasonal dynamics), which also lends itself to support the reasoning for why it is a strong driver of plant's elevation distributions in a more ecological-evolutionary framing.

Main Text Line-specific comments:

L190 Without more clarity on the specific assumed interactions between within and among generation variation in the main text, it is not clear to me how this lack of support for the Gilchrist hypothesis fits into the other hypothesis and analyses in this work. This section is very unclear to me, and also may benefit from a larger focus on the multiple scales of variability examined for the main analysis of this work.

Methods Line-specific comments:

L72 "We truncated species' elevational ranges crossing the lower end of the standardized elevational gradients, keeping the parts of the species' ranges within the gradient's

boundaries only.” Wouldn’t this artificially reduce the range size of lower elevation species – this seems problematic for both the global and mountain-level analyses. Was this done so that it is more similar to the high elevation limit of species that is truncated by the mountain peak elevation?

Reviewer #3 (Remarks to the Author):

This study challenges the application of one of the major rules in ecology, i.e., Rapoport’s rule, coming with an opposite conclusion. It is therefore intriguing and insightful. I have, however, a series of questions regarding the data and the interpretation of the results. There is an increasing awareness that organisms experience climatic conditions that may substantially differ from those recorded by meteorological weather stations, hence an increasing interest for fine-scale climatic data. This is especially true in mountain areas, where climatic conditions vary substantially across short distances—and therefore, even the wide scale of the present analysis does not prevent from taking this issue into account. This raises an issue that is not addressed here, which is the spatial resolution of the data. This includes the species distribution data and the climatic data. Regarding species data, the data that were used were filtered following different criteria, but nothing is said about the spatial resolution of those data. Information regarding this would be necessary, and if no criterion was implemented to filter-out data with a resolution of less than, at least, 250m, I suggest that the analyses focus on high-resolution distribution data.

Regarding the climatic data, it appears that the data were downloaded from Chelsa at a spatial resolution of 30”. As suggested above, there can be a huge variation in mountain regions among sites separated by 1km—a range difference of 1000m is actually larger than the range that some species can actually occupy. There is a mention of 100m elevation bands in the M&M section, but if climatic data are at 30” resolution, how was it possible to compute ‘the mean bioclimate value for each mountain in 100 m elevation bands’? By interpolating the data available at a 1km resolution? If so, using what model? Even though the climatic data were properly downscaled, how was climatic variation of one band characterized? By computing a mean across all 1km² pixels that are constitutive of the corresponding 100m elevation layer in the considered mountain system?

The M&M section would need to be substantially expanded so that the reader can

understand what was actually done. I have even a basic question, to which I did not find the answer in that section: the analysis is based, for each species, on its elevation range and the daily temperature variation that it experiences. If a species occurs across several 100m elevation bands, how was that daily temperature variation characterized? By computing an average across all occupied elevation bands?

Regarding the interpretation of the results, it seems to me that, as opposed to Rapoport's rule, Alpine species have actually narrower ranges than low elevation ones. This, however, has nothing to do with a thermal range: most Alpine species thrive at low elevation, there are excluded from the latter in the wild due to competition with low elevation species. This suggests that while the pattern described here might well be correct, the underlying mechanism may be challenged: in fact, the diurnal variation of temperature is actually highly correlated to elevation, which itself is a compound variable associated with many factors characterizing environmental differences along an elevation gradient. Thus, I wonder whether daily temperature variation per se drives the observed pattern, or if it is any other variable correlated with the elevation gradient.

Finally, I wonder how to reconcile the present results with all previous empirical studies lending support to Rapoport's rule, and in particular, papers such as Sandel et al. 2011 (Science) demonstrating that species ranges are linked to the velocity of climate change, showing that, at opposed to what is presented here, species with narrow climatic ranges went extinct during glacial events, so that species wide ranges are characterized by large climatic tolerances? A discussion of the discrepancies between the present findings and the (large number of) studies supporting Rapoport's rule would be insightful.

We are grateful to the reviewers for their constructive criticism and comments, which we believe have significantly strengthened the manuscript. We have adjusted the text and figures accordingly. Detailed replies are presented below. We have also reorganised the figures in the appendix to maintain one figure per page, which makes the figures easier to read.

Reviewer #1

This is very nice study, clearly presented and convincing. The findings indicate that, in contrast to previous ideas, elevation range sizes of species are smaller when there is higher temporal (especially diurnal) variation. I do not have substantial comments, only two major things which might be useful to consider. First, trends of range sizes along gradients (e.g. latitudinal or altitudinal) are subjects to several geometrical effects which distort the patterns. Most importantly, domain boundaries may truncate potential ranges given by climatic tolerance, so that observed range size patterns differ from the patterns comprising potential ranges. In this respect, it may be useful to consider paper of Sizling et al. (2009; Ecology 90: 3575-3586) which explicitly treats these issues (albeit on a latitudinal gradient, but the principle is universal).

Thank you for your enthusiasm and for pointing out the Sizling et al. paper. This paper nicely illustrates some of the problematic issues with many of the previous papers on the geographical distribution of range sizes, and we have added it to the reference list in the method section. We added Extended Data Fig. 8 to illustrate the influence of standardization of elevation spans. The new figure shows that there is a clear relationship between non-standardized range sizes and full elevation spans (at least when including full elevation spans < 2500 m), but that this relationship disappears after standardizing the gradient lengths. The figure also shows that standardization accounts for the differences between islands and continental mountains.

I acknowledge the authors have been aware about these issues and tried to avoid the effect of range truncation by standardizing elevation ranges and not counting species whose ranges are located just next to the domain boundary. Still, it is not clear if these measures completely remove these effects. Since I do not have any better recipe than what has been done by the authors, I would like at least to see whether there is an effect of the length of elevation gradient on the range size, and/or map of the locations with the

indication of the lengths of the gradients (before standardization, of course) and mean range size (in addition to the map which is already presented in Extended data). For instance, are tropical localities those with longer elevation gradients and at the same time shorter ranges? I still think that shorter elevation gradients may shrink species ranges, even if the gradients are standardized by using only the upper part of the gradient with the standard size.

We added Extended Data Fig. 8 to illustrate the influence of standardization of elevation spans. The new figure shows that there is a clear relationship between non-standardized range sizes and full elevation spans (at least when including full elevation spans < 2500 m), but that this relationship disappears after standardizing the gradient lengths. The figure also shows that standardization accounts for the differences between islands and continental mountains. This relationship is negative when not differentiating island and continental mountains (i.e. opposite of what we would expect if this was an issue) since islands have in average larger range sizes, probably caused by the differences in short-term temperature variation.

Second, the authors seem to assume that the range sizes are determined by limits of temperature tolerance. But it seems to me that temperature may limit only the upper range edge, whereas the low-elevation edge is often determined by interspecific competition and interactions generally (including natural enemies, parasites etc.) – species often can potentially live in lower elevations, but they are outcompeted by better adapted species there. How this go together with the reported patterns?

In the model we developed temperature variation will theoretically limit range sizes in both directions (lower and upper ends, as shown in Fig. 1). In practice, we agree that, within individual mountains, the influence of temperature variation in limiting species' range sizes can be asymmetrical. E.g. temperature variation can have a stronger influence toward colder habitats (higher elevations) than at lower elevations. We also acknowledge the existence of many factors that can conjointly explain plant distribution patterns in the main text on lines 180-181. Most importantly, the conclusions of the study remain the same even in the case of an asymmetrical influence of temperature variation on species' range sizes. This being said, the purpose of the paper is to explore the theoretical concepts of the temperature range squeeze hypothesis. This is why we focused on the potential influence of temperature variation per se in shaping species distribution, more than exploring the plethora of factors influencing species ranges.

Also, as far as I understand, daily variation has been estimated for the whole mountain(s) as the data unit. But daily temperature variation changes along elevation gradients, so that range limits in different elevations are driven by different factors. Again, at the moment I do not have an idea how to resolve these issues, but I just think there may be effects not considered by the authors, although I appreciate the way how the authors have dealt with some confounding effects.

We agree that daily temperature variation will vary along elevation gradients within mountains. However, the variation is also generally much smaller than the variation we observed across mountains, and would therefore have minor influence on the global analyses. Please compare the range of temperature variation in Fig. 2 (about 12°C for diurnal temperature range) to the range of thermal variability within mountains in Extended Data Fig. 12 (about 2°C for the elevation gradient with the greatest within-mountain variation in diurnal temperature range).

Minor comments:

Fig. 2: Continental mountains seem to be marked by black, not gray circles.

We replaced “gray” with “dark gray” to be more precise.

Line 119: Should it sound ‘...maybe lower.’ rather than ‘...may be less’?

Replaced with “may be lower”.

Methods, lines 6-7: This repeats what has been already in the main text.

We removed “defined as mountainous areas, countries, islands, mountain ranges or volcanoes” from the Methods section.

Reviewer #2

This work uses a global dataset of plant distributions in mountains to examine the relationships between diurnal, seasonal, and longer-term temperature variability and plant elevation range size, and finds a strong relationship between elevation range size and diur-

nal temperature variation. The plant distributional data collated for this work was impressive, and the analyses were well-explained and statistically robust. My major criticism of this work was the framing, which sets up the longstanding, but often criticized, theory for a positive variability-range size relationships (Janzen/Steven) against a newly presented Temperature Squeeze Hypothesis. Although this new hypothesis is well-explained with Figure 1, the comparison between the only these two hypotheses seemed limited in terms of the large number of potential drivers of plant range sizes (Sheth et al. 2020 New Phytologist).

We acknowledge the fact that many factors come into play in shaping species' range sizes in the main text (L180-181). However, the aim of the present study was to investigate the influence of temperature variation on species' elevation ranges and propose an alternative and logical explanation to previous hypotheses linking range size to thermal variability. Hence the primary focus on temperature variation only. We did not aim at investigating the influence of all potential drivers of species' elevation ranges.

Secondly, I was struck by the lack of exploration of the different processes (evolutionary vs. ecological) that are largely invoked for driving these two hypothesized patterns, with the former focused on the evolved response of thermal tolerance and the latter focused on ecological limitation through abiotic filtering of species with set thermal tolerances. It is well recognized that range limits are likely driven jointly by evolutionary and ecological dynamics (Sexton et al. 2009), and this possibility it noted by the authors implicitly in L77-79. The current framing of the two hypotheses doesn't make it clear enough that one hypothesis is more of an evolutionary argument while the other is more ecological where thermal niches are assumed to be a stationary species-level characteristic. I don't think that this adjustment of the framing would reduce the novelty of this work or lack of support for the findings. Range limit theory in papers already cited in the work present many reasons (low population size, gene swamping etc.) why range edge populations may not necessarily be able to adapt to the thermal extremes experienced in those populations (even if considering niche limits not to be lethal), and therefore abiotic filtering may be a larger driver of range sizes in mountains.

We added a sentence to be more explicit about the fact that Stevens's hypothesis is evolutionary focused while our new hypothesis is ecologically focused (L81-84 in the main text).

Further, the finding of the relationship between diurnal temperature variability and elevation range size is an exciting one, and I think that the introduction and work overall could have a stronger focus on these multiple scales of variability explored, the predictions for relationships between variability and range size at these different scales, and well as the implication for the patterns found. The authors argue how this type of variability is difficult to compensate for both through avoidance (e.g., dormancy) and adaptation (as it may be less predictive than seasonal dynamics), which also lends itself to support the reasoning for why it is a strong driver of plant's elevation distributions in a more ecological-evolutionary framing.

We thank the reviewer for suggesting that we should emphasise this aspect of the work and have added further details in the introduction (L96-104).

Main Text Line-specific comments: L190 Without more clarity on the specific assumed interactions between within and among generation variation in the main text, it is not clear to me how this lack of support for the Gilchrist hypothesis fits into the other hypothesis and analyses in this work. This section is very unclear to me, and also may benefit from a larger focus on the multiple scales of variability examined for the main analysis of this work.

In addition to the expansion of text focusing on explaining the multiple scales of variability examined (see previous point) we have added more details about the interactions assumed by Gilchrist in the main text on lines 146-149.

Methods Line-specific comments: L72 "We truncated species' elevation ranges crossing the lower end of the standardized elevation gradients, keeping the parts of the species' ranges within the gradient's boundaries only." Wouldn't this artificially reduce the range size of lower elevation species – this seems problematic for both the global and mountain-level analyses. Was this done so that it is more similar to the high elevation limit of species that is truncated by the mountain peak elevation?

Yes, it artificially reduces the range size of lower elevation species just like the bottom (or top) of a mountain truncates species range sizes due to physical boundaries. We did this to make species' ranges comparable across different elevation spans and cancel the bias created by the otherwise positive relationship between species' ranges and mountain height. The rationale

between standardizing elevation gradients from upper or lower elevations is debatable. One could argue that standardizing from mountain tops can help reduce the influence of human activities and urbanization at lower elevations, in particular in higher mountains since lower elevations would simply be excluded from the analyses. However, since there's no good answer to this question, we also performed the analyses by standardizing elevation gradients from lower elevations (Extended Data Fig. 2), which yielded similar results.

Reviewer #3

This study challenges the application of one of the major rules in ecology, i.e., Rapoport's rule, coming with an opposite conclusion. It is therefore intriguing and insightful. I have, however, a series of questions regarding the data and the interpretation of the results. There is an increasing awareness that organisms experience climatic conditions that may substantially differ from those recorded by meteorological weather stations, hence an increasing interest for fine-scale climatic data. This is especially true in mountain areas, where climatic conditions vary substantially across short distances—and therefore, even the wide scale of the present analysis does not prevent from taking this issue into account. This raises an issue that is not addressed here, which is the spatial resolution of the data. Regarding species data, the data that were used were filtered following different criteria, but nothing is said about the spatial resolution of those data. Information regarding this would be necessary, and if no criterion was implemented to filter-out data with a resolution of less than, at least, 250m, I suggest that the analyses focus on high-resolution distribution data.

Species data come from various sources (most commonly checklists with expert estimates of species ranges but also plot samples and human observations) that are not gridded and with no explicit mention about the precision of GPS coordinates used during the sampling. Data sets have a resolution < 100 m along the elevation gradient. We added that information in the Methods on line 7.

Regarding the climatic data, it appears that the data were downloaded from Chelsa at a spatial resolution of 30". As suggested above, there can be a huge variation in mountain regions among sites separated by 1km—a range difference of 1000m is actually larger

than the range that some species can actually occupy. There is a mention of 100m elevation bands in the M&M section, but if climatic data are at 30" resolution, how was it possible to compute 'the mean bioclimate value for each mountain in 100 m elevation bands'? By interpolating the data available at a 1km resolution? If so, using what model?

The global-scale analyses (across mountains), which are the main analyses of the paper, estimate and average temperature variation (daily, seasonal or $\Delta MAT0-1980$) in the entire mountain region (i.e. across all elevation bands). The local-scale analyses (within-mountain) average values from all cells falling inside every given elevation band in each mountain. For these estimates we do not expect the relative differences between mountains (or elevation bands) to be dependent on using micro- or macro-climatic variables. We don't believe that using climate at a micro scale is a better option in general as species' ranges are estimated for the entire mountain range, making it hard to correlate species' range sizes to microclimate, which is restricted to specific areas. Differences between microclimate and low resolution climatic data may introduce an over- or underestimation of temperature variation in some places but such biases shouldn't be systematic since we are working across the full elevation gradient of each species and location. Species data are mostly given for entire mountains/areas, not for specific locations within each mountain. Linking such species data to micro-climate means we'd have to average micro-climate to the entire mountain which we believe would be very similar to using lower scale bioclimate data as we did.

Even though the climatic data were properly downscaled, how was climatic variation of one band characterized? By computing a mean across all 1km² pixels that are constitutive of the corresponding 100m elevation layer in the considered mountain system?

Yes, this is what we did. We rephrased the sentence describing how we computed climate data within elevation bands to make it more clear (lines 35-37 in the Methods).

The M&M section would need to be substantially expanded so that the reader can understand what was actually done. I have even a basic question, to which I did not find the answer in that section: the analysis is based, for each species, on its elevation range and the daily temperature variation that it experiences. If a species occurs across several 100m elevation bands, how was that daily temperature variation characterized? By computing an average across all occupied elevation bands?

We reformulated the description of the main analyses and hope it has improved the clarity of the analyses. We tried to clarify that the global analyses are based on using averaged range sizes within mountains, which are then related to the averaged bioclimate per mountain. We also tried to make a clearer distinction between the global-scale analyses and the local scale analyses in the introduction of the description of the statistical analyses. We hope it makes it easier to understand each approach and their respective differences. Specifically for the local-scale analyses, we assigned climate data to species' range midpoint as mentioned in the Methods on line 114. Please see Rohde et al. (Rapoport's Rule Does Not Apply to Marine Teleosts and Cannot Explain Latitudinal Gradients in Species Richness; 1993) for a discussion on this approach.

Regarding the interpretation of the results, it seems to me that, as opposed to Rapoport's rule, Alpine species have actually narrower ranges than low elevation ones. This, however, has nothing to do with a thermal range: most Alpine species thrive at low elevation, there are excluded from the latter in the wild due to competition with low elevation species.

We are not sure if we agree with the interpretation of our results. The local-scale analyses (which could support the hypothesis that alpine species have narrower ranges) do not show a consistent pattern (Fig. 4) as there's the same amount of mountains with positive and negative relationships. Our main analyses are the global analyses which are not comparing alpine species with low-elevation ones, but rather merge the two sets of species and take the average of all species (independent of if they are alpine or lowland species) per mountain.

This suggests that while the pattern described here might well be correct, the underlying mechanism may be challenged: in fact, the diurnal variation of temperature is actually highly correlated to elevation, which itself is a compound variable associated with many factors characterizing environmental differences along an elevation gradient. Thus, I wonder whether daily temperature variation per se drives the observed pattern, or if it is any other variable correlated with the elevation gradient.

This is mostly true for the local-scale analyses. In fact, this point is part of our explanation on why we didn't find any trends in the local-scale analyses (L180-181 in the main text). In the global-scale analyses, we averaged range sizes and bioclimates to entire mountains and looked at the relationship between species' ranges and temperature variation across mountains. Doing so helps us decouple climatic and geographic factors, making the interpretation of the results

easier. We changed the text to make this point clearer on lines 57-59 in the Methods. On a side note, we didn't find strong evidence of correlation between diurnal temperature variation and elevation. Diurnal temperature variation is essentially correlated to drought. It can be positively or negatively correlated to elevation depending on whether the driest areas are located at lower or upper elevations.

Finally, I wonder how to reconcile the present results with all previous empirical studies lending support to Rapoport's rule, and in particular, papers such as Sandel et al. 2011 (Science) demonstrating that species ranges are linked to the velocity of climate change, showing that, at opposed to what is presented here, species with narrow climatic ranges went extinct during glacial events, so that species wide ranges are characterized by large climatic tolerances?

We're aware that many studies support Rapoport rule. There are also studies that don't support it (please see references 3-7 in the main text). We used a different approach that we believe can explain why we got different results than papers showing support for Rapoport rule:

1. Most previous studies used elevation or latitude as predictor variables for range sizes and subsequently interpreted this as a proxy for temperature variation, when we looked at the influence of temperature variation on species' range sizes directly. In our study we also found that diurnal temperature range is poorly correlated with elevation/latitude, meaning that the species' range-diurnal temperature variation relationship couldn't be derived from past works using elevation/latitude as the explanatory variable.
2. Past works investigating the relationship between species' elevation range and temperature variation essentially performed their analyses along single elevation gradients. We performed similar analyses ("local-scale analyses") that resulted in as many positive as negative relationships, which would tend to align with the many contradictory results about Rapoport rule found in the literature. Only a handful of papers analysed species' range sizes across mountains as we did in our global-scale analyses, and all of these used organisms other than plants.
3. The vast majority of previous papers analysed data without considering the influence of mountain heights and the proximity of species to physical barriers (please see comments and reply to Reviewer #1). We showed in Extended Data Fig. 13 that the bigger the exclusion zone, the more uncertain the relationships. In other words, at a local-scale, many

of the patterns seem to be driven by those species found exclusively next to the domain edges (upper-/lower-most species) that are likely to have truncated range sizes.

REVIEWER COMMENTS

Reviewer #1 (Remarks to the Author):

All my major concerns have been addressed. While I am still not entirely convinced that all potential artefact given by range truncations have been addressed and/or avoided, I do not have any recommendation how to deal with these issues other than the ways used by the authors, so I am happy with the current version of the manuscript.

Reviewer #2 (Remarks to the Author):

Thank you to the authors for your considerate and careful responses to the reviewer comments - my previous comments have all been adequately addressed or explained.

In reviewing the full manuscripts again, I think there is a typo in the caption of Extended Data Fig. 4. I believe the captions for Extended Data Fig. 4 and Extended Data Fig. 3 are identical, and that Extended Figure 4 should be showing data for a standardized elevation span of 2000m

Reviewer #3 (Remarks to the Author):

The manuscript has been superficially revised. I still have major questions remaining, mostly because the methodology is insufficiently detailed.

I am still very confused about the climatic data used and the issue of spatial resolution. The M&M section indicates that: 'We computed mean bioclimate values within each 100 m elevation band in every mountain'. This means that daily temperature range must have been obtained for 100m elevation bands, and then averaged across bands. The spatial resolution of the climate data is, however, 1km: —hence the question, which remains unanswered in the M&Ms: how were these temperatures obtained for a band of 100m from data that have 1km resolution?

From what I understand, these daily temperature range values (thus obtained by interpolation, using elevation as a predictor, otherwise I do not understand how they were generated) were averaged across the entire elevation gradient, and again averaged across

the constitutive pixels belonging to the same 100m elevation band (across quite large areas, as sometimes a 'mountain' is a country). I am not sure what ecological information results from an average of average of interpolated data, but for sure, this information does not reflect the actual daily temperature range experienced by species. I would suggest to add a map, showing the spatial distribution of these values across the study areas. I would think that these values would potentially follow a latitudinal pattern (as one expects day temperature variation to decrease towards the tropics). If this is the case, this suggests that species elevation ranges correlate with latitude, a pattern evidenced, among others, by McCain (2009 *Ecol. Lett.*).

I understand that species elevation ranges themselves were averaged across species per study area. Once again, I am not sure what this represents and how to interpret the spatial variation of such a value, and a map showing the spatial distribution of these average elevation ranges across species would be useful to help interpreting the patterns. I further wonder whether in Fig2, error bars represent the standard deviation of the elevation range across species. In fact, the bars are extremely small: if they represent an actual st.dev. of the elevation ranges across all species of a mountain range, one would expect this st.dev. to be of several hundreds of meters, reflecting the fact that there is a mixture of species with narrow and large elevation ranges.

This further raises the question of how those elevation ranges were inferred. Species data seem to correspond to casual observations, and this raises the question of whether these observations were performed across the actual elevation range of the species. There is thus a need to show that (i) these data were collected across the entire elevation range of those species, and do not represent a subset of it based on a series of random observations, and (ii) the spatial resolution of those data was sufficient. There is no information regarding this in the ms, whereas this is a key issue: if the spatial resolution of the data is of a few hundreds of meters, this challenges the idea that they can be used to characterize an elevation range of 500-1000m.

All in all, this suggests to me that the day temperature range is an intriguing hypothesis, but that it should remain as such: the ms illustrates a pattern that is consistent with it, but that does not evidence it: as the rebuttal letter makes it explicit, many other mechanisms than day temperature range could be involved. There is one sentence acknowledging this in the ms, but in my opinion, this is not sufficient: the ms reads as if day temperature IS the causal

factor, and the title reinforces this. To me, the title should be changed and the ms substantially revised to strongly de-emphasize the role that day temperature may play: of course, the ms will be much less conclusive, but it will be a closer reflection of the reality. It seems to me that the present version is over-conclusive. A macro-ecological pattern is revealed, what are its underlying mechanisms is far from clear.

We are grateful to the reviewers for going through the manuscript one more time and are glad that the reviewers feel that most of the issues raised have been addressed. We have addressed the remaining issues, and believe that this has further improved the manuscript.

Reviewer #1

All my major concerns have been addressed. While I am still not entirely convinced that all potential artefact given by range truncations have been addressed and/or avoided, I do not have any recommendation how to deal with these issues other than the ways used by the authors, so I am happy with the current version of the manuscript.

We thank Reviewer 1 for his/her valuable suggestions and are happy to see that the reviewer considers that all major concerns are addressed in this current version.

Reviewer #2

Thank you to the authors for your considerate and careful responses to the reviewer comments - my previous comments have all been adequately addressed or explained.

We are thankful to Reviewer 2 for the many useful comments and for the positive feedback.

In reviewing the full manuscripts again, I think there is a typo in the caption of Extended Data Fig. 4. I believe the captions for Extended Data Fig. 4 and Extended Data Fig. 3 are identical, and that Extended Figure 4 should be showing data for a standardized elevation span of 2000m

Thank you for catching this mistake. We fixed the captions.

Reviewer #3

The manuscript has been superficially revised. I still have major questions remaining, mostly because the methodology is insufficiently detailed. I am still very confused about the climatic data used and the issue of spatial resolution. The M&M section indicates that: 'We computed mean bioclimate values within each 100 m elevation band in every mountain'. This means that daily temperature range must have been obtained for 100m elevation bands, and then averaged across bands. The spatial resolution of the climate data is, however, 1km: —hence the question, which remains unanswered in the M&Ms: how were these temperatures obtained for a band of 100m from data that have 1km resolution? From what I understand, these daily temperature range values (thus obtained by interpolation, using elevation as a predictor, otherwise I do not understand how they were generated) were averaged across the entire elevation gradient, and again averaged across the constitutive pixels belonging to the same 100m elevation band (across quite large areas, as sometimes a 'mountain' is a country).

We are not sure that we fully understand this criticism, given that 100 m elevational bands are in the vertical dimension while the climate data are in the two horizontal dimensions. We think the criticism (exemplified by the question “how were these temperatures obtained for a band of 100 m from data that have 1 km resolution?”) must relate to confusion between vertical and horizontal distances; we note that 100 m elevational bands typically (though not always) span more than 1 km horizontally. In case this is confusing also to other readers, in the revised manuscript we expanded the explanation of elevational bands in lines 243-249. We hope it is now crystal clear. We resampled climate rasters to the resolution of SRTM rasters (1 arc-second) to compute mean climate values by elevation band. The resampling consisted of subdividing climate cells into smaller cells so that we could match climate values with elevation values **without interpolating** climate data to a higher resolution. Although a higher climate resolution would have been ideal, our approach allows us to capture enough variation in temperature along elevation gradients to perform our analyses. Besides, we selected mountain regions spanning less than 1000 km on a North-South axis to limit variation in temperature related to latitude, as mentioned in the Methods in line 214.

I understand that species elevation ranges themselves were averaged across species per study area. Once again, I am not sure what this represents and how to interpret the spatial variation of such a value, and a map showing the spatial distribution of these average elevation ranges across species would be useful to help interpreting the patterns.

We thank the reviewer for this question and suggestion. To clarify this better and support our approach, we added a new map (Supplementary Fig. 2) in the appendix showing mean species' range sizes in the 44 mountains used in the study.

I am not sure what ecological information results from an average of average of interpolated data, but for sure, this information does not reflect the actual daily temperature range experienced by species. I would suggest to add a map, showing the spatial distribution of these values across the study areas. I would think that these values would potentially follow a latitudinal pattern (as one expects day temperature variation to decrease towards the tropics). If this is the case, this suggests that species elevation ranges correlate with latitude, a pattern evidenced, among others, by McCain (2009 *Ecol. Lett.*).

We thank the reviewer for bringing this up as it gives us more opportunity to expand on our reasoning of using DTR instead of latitude.

The spatial variation in the climatic variables asked for can be seen in Chan et al. 2016 (see Fig. 3). The maps show the spatial distribution of diurnal temperature range (DTR), temperature seasonality and mean annual precipitation as well as the combination of all three variables. Unlike temperature seasonality, DTR is usually not directly correlated with latitude, but closely linked to precipitation/air moisture, cloud cover and sunshine duration (see e.g. Shen et al. 2014 or He et al. 2015).

In addition we would like to avoid using latitude as a predictor variable. Latitude per se has no ecological meaning, but is associated with variables of ecological interest (e.g. mean annual temperature), which is why we did not focus on it in the study. However, to satisfy the reviewer's curiosity, below is a scatter plot with a linear regression of mean elevation range sizes vs latitude from our data. As the plot shows, our data do not support the idea that species' range sizes strongly correlate with latitude.

I further wonder whether in Fig2, error bars represent the standard deviation of the elevation range across species. In fact, the bars are extremely small: if they represent an actual st.dev. of the elevation ranges across all species of a mountain range, one would expect this st.dev. to be of several hundreds of meters, reflecting the fact that there is a mixture of species with narrow and large elevation ranges.

Bars in Fig. 2 (and other scatter plots) are not standard deviations of elevation ranges. Instead, they show standard errors, as mentioned in the figure caption: *"Points represent the estimated mean elevation ranges with their respective standard error"*.

This further raises the question of how those elevation ranges were inferred. Species data seem to correspond to casual observations, and this raises the question of whether these observations were performed across the actual elevation range of the species. There is thus a need to show that (i) these data were collected across the entire elevation range of those species, and do not represent a subset of it based on a series of random observations

It's unfortunately impossible to know what the exact full (elevation) ranges of species are across all mountains as range sizes are always an (under-) estimation of actual species' ranges. The difficulty in getting exact species' distribution ranges lies in the impossibility of sampling all individuals, even with exhaustive sampling conducted over a long period of time. This is often

due to study areas that are too large/remote to be fully sampled or simply because of individuals that are not recorded. Not knowing the exact full range size of all species is a recurrent issue in studies on the geographic distribution of species.

We do want to emphasize, however, that over the course of the last century, long-standing hypotheses have been brought forward relying on much less rigorous and extensive data sets and statistical approaches than ours, and numerous follow-up studies to validate or disprove these hypotheses have been published using data sets that faced the same issues.

To support our analysis with the best data set currently possible, we covered all matters of:

- *Multiple data sources:* We used multiple sources of data to compile our species observations, including field surveys, scientific literature, and reliable species databases. This approach allowed us to gather information from a wide range of studies and observations conducted by various researchers, increasing the overall coverage of the species' elevation ranges.
- *Data validation and quality control:* We implemented rigorous data validation and quality control measures to ensure the reliability of the species observations. We cross-checked the information from different sources and verified the accuracy of elevation data whenever possible. Any inconsistent or unreliable data points were carefully identified and excluded from the analysis to limit potential errors or biases. E.g. excluding multiple mountains that had heterogeneous sampling along the elevation gradient and a high proportion of single observations.
- *Expert knowledge:* Our research team consisted of experts in the field of species distribution and mountain/island ecology. Our work is supported by specialists who have extensive knowledge of the study areas included to validate the collected data and ensure that it represents a comprehensive sample across the species' elevation ranges.
- *Statistical modeling:* We employed advanced statistical modeling techniques to estimate species range sizes within each mountain. These models take into account the available data and provide a quantitative estimate of the species' distribution patterns, considering the elevational gradient. By incorporating the modeling approach, we aimed to mitigate potential sampling issues and provide more accurate estimates of the species' elevation ranges.
- *Sensitivity analyses:* To assess the robustness of our results, we conducted sensitivity analyses by varying the inclusion/exclusion criteria and evaluating the impact on the final

outcomes. This allowed us to examine the influence of different factors and criteria on the observed patterns and confirm the consistency and reliability of our findings.

In conclusion, we believe that we have taken appropriate measures to ensure the reliability and representativeness of the data used in our study, tackling the issues around elevational range estimates in the best way currently possible.

and (ii) the spatial resolution of those data was sufficient. There is no information regarding this in the ms, whereas this is a key issue: if the spatial resolution of the data is of a few hundreds of meters, this challenges the idea that they can be used to characterize an elevation range of 500-1000m.

We had already foreseen that this could be an issue and therefore we circumvented this in our approach. We point the reviewer to the lines where we address this matter; one of the criteria for including a mountain in the final analyses was that the elevational resolution of species data is less or equal to 100 m as mentioned in the Methods in line 215: “[...] *elevation range size data were given explicitly for the relevant mountain areas with an elevational precision \leq 100 m*”. We also refer the reviewer to our previous comment regarding the confusion around 100 m elevation bands across horizontal and vertical distances, to support the notion that elevation ranges of 500-1000 m will be well captured by the horizontal resolution of our used SRTM.

All in all, this suggests to me that the day temperature range is an intriguing hypothesis, but that it should remain as such: the ms illustrates a pattern that is consistent with it, but that does not evidence it: as the rebuttal letter makes it explicit, many other mechanisms than day temperature range could be involved. There is one sentence acknowledging this in the ms, but in my opinion, this is not sufficient: the ms reads as if day temperature IS the causal factor, and the title reinforces this. To me, the title should be changed and the ms substantially revised to strongly de-emphasize the role that day temperature may play: of course, the ms will be much less conclusive, but it will be a closer reflection of the reality. It seems to me that the present version is over-conclusive. A macro-ecological pattern is revealed, what are its underlying mechanisms is far from clear.

We agree that our results on the influence of DTR on species' elevation ranges do not mean that other factors are irrelevant or have no influence, we are explicit about this point in the text in line 181-183 and 197-198.

We acknowledge that on the local scale our hypothesis is not the overruling driving factor, but on a global scale our patterns are consistent enough to question the long-standing hypothesis by Stevens. We present a strong case - both theoretical and empirical - against the commonly accepted idea that 1) seasonal variation is the main driver in shaping species' elevation ranges and 2) species' range sizes broaden when seasonal variations increase. Therefore, we believe that our paper provides an innovative and fresh perspective in the discussion on elevational ranges, and completely downtoning the relevance of DTR would not do justice to these important results.

To follow up on the reviewer's concerns, we did, however, adjust the title to "*Diurnal temperature range as a key predictor of plants' elevation ranges globally*" and we adjusted the final part of the paper as follows:

In summary, our findings solidify the novel temperature range squeeze hypothesis (Fig. 1c-d) that predicts declines in species' range sizes in climatically variable habitats, thereby contradicting Stevens' hypothesis (Fig. 1b). Our research suggests that diurnal variation in temperatures plays a determining role in shaping the elevation ranges of vascular plants on a global scale and contributes to the differences in range sizes observed between continents and islands. Local (within-mountain) variation in species' elevation ranges, however, is largely decoupled from temperature variability and could be the result of local, interacting variables, such as species interactions, land use and soil type, among others. The detected global signal of the role of diurnal variation in temperature in shaping elevation ranges urges the need to reconsider past theories on our understanding of the driver of plant species distributions. Especially for the field of conservation biology in the face of global change, these novel insights are important to take into account. Our findings call into question the prevailing understanding that global changes will especially imperil tropical species with restricted ranges. We suggest that extinction risk may be higher in continental mountains where species are more likely to have smaller elevation ranges due to higher (diurnal) thermal variability than species inhabiting mountain systems exposed to oceanic climates. Short-term temperature variation may also become increasingly important in driving local extinction risks but a large variation of local variables will need to be considered. Considering our findings, the temperature range squeeze hypothesis holds true on a global scale, shedding light on the determining influence of diurnal temperature variation on species' elevation ranges and emphasizing the need for adaptive conservation measures in the context of a rapidly changing climate.

REVIEWERS' COMMENTS

Reviewer #3 (Remarks to the Author):

Thank you for the revision of the ms and additional explanations. I still have, however, the same major concerns. If I understood correctly, the climatic data of all the 1km² pixels partly overlapping a 100m elevation band were used to characterize the climatic conditions of that band. My point is that climatic data at 1km resolution in a mountain area are extremely crude and not representative of the conditions actually experienced by the species (see eg <https://doi.org/10.1111/ecog.03947>, <https://doi.org/10.1111/gcb.12129>) I read in the rebuttal letter 'that 100 m elevational bands are in the vertical dimension while the climate data are in the two horizontal dimensions'. I either do not understand this answer or disagree with it: within a 1 km² pixel in a mountain region, elevation can of course vary drastically, involving huge variations in climatic conditions at two sampling points located at several hundreds of meters of each other (horizontally) but at different elevation within the same pixel. In short: Chelsa data for a 1km² pixel in a mountain region crudely reflect the climatic conditions that can vary across several hundreds of meters. As Meinari & Hylander (doi:<https://doi.org/10.1111/ecog.02494>) noticed, 'Coarse-grain climate data are often more representative of the lower altitudes than the higher altitudes encountered within each pixel' and 'the thermal tolerances of species (when using coarse-grain data) may be overestimated. This arises when a species occurs somewhere in a large grid cell even though the cell's mean climate is outside the species' climatic tolerance limits'. I therefore still challenge the idea that one can characterize species diurnal temperature range with the data at hand.

The present study reveals, however, a clear pattern, as evidenced in Fig2. I nonetheless question the fact that elevation range is driven by diurnal temperature range, given all the limitations in extracting the latter from data that do not have the resolution that would be required to address this question. As shown by the graph in the cover letter, latitude alone is not either the main factor for the observed pattern... but is, for sure, part of the explanation, as one can see a clear decrease in the mean elevation range with latitude, as previously evidenced by McCain (2009). All in all, I like the topic of the study and the wide spatial scale at which it was performed. A large spatial scale does not mean, however, that data at crude resolution can be used. Due to this issue of resolution, but also due to the fact

that a very limited number of factors were analyzed, I do not believe that the study provides evidence for the role of diurnal temperature range in explaining elevation range. I suggest that elevation range is determined by a series of factors, which should be sampled at fine resolution to accurately describe species temperature ranges, whose contribution should be assessed in a multivariate context.

Reviewer #4 (Remarks to the Author):

I got 'called in' as an additional expert to weigh in on the remaining issue in the underlying paper, regarding the question of the climate data at hand is suitable to answer the research question posed here.

I will start by saying that I really liked the paper, that it was very easy to read, and very elegantly written. It is always nice to see papers aimed at confirming or disproving general ecological theory, so I applaud the authors for their attempt!

However, the debate between authors and reviewer is an important one, and I hope I can contribute with some constructive thoughts:

- I follow reviewer 3 that CHELSA climate data doesn't give a very good proxy of local conditions as experienced by plants in the mountains, due to its coarse resolution and it only being relevant for free-air conditions and not in-situ temperatures. This is especially relevant actually for diurnal temperature ranges, which are strongly affected by the buffering of 1) topography and 2) vegetation. By using the spatial heterogeneity in diurnal temperature ranges, which can be huge in mountain areas, plants can avoid many of the coarse-resolution 'lethal temperatures'. A good paper in that regard is Maclean and Early 2023, which provides a neat theoretical figure on how species can live in particular fine-grained pixels within a coarse-grained pixel that is in theory unsuitable. Of course, this doesn't mean that overarching macroclimatic patterns cannot be detected – we have been doing macro-ecology with macroclimate data successfully for years – but it is at least an important discussion point to add to the paper.

- Related to the above, the actual experienced diurnal pattern would be significantly different for forest understory versus open area species. And as long as the buffering of forests is the same everywhere, your macroclimate relationship would hold up. However, this is not the case, a similar forest cover could be buffering temperatures more in a wet

than in a dry climate, for example. However, I don't think this would necessarily affect the observed patterns.

- To me, something interesting and important is going on in Fig. 2a: the strong pattern in relationship with diurnal temperature range seems largely driven by the island vs mainland dichotomy. Indeed, within mainland sites, the relationship with diurnal temperature range is significantly weaker. This in itself is possible, but the authors state 'diurnal temperature range was the only variable to account for variation in species' range sizes between island and continental mountains.' (L156-157) I think this should be nuanced to 'the only variable that we tested for'. As I can imagine, in line also with reviewer 3's thoughts, I believe, that other parameters (for example continentality, to name just one, or simply distance to the ocean) would also account for the variation in species' range sizes between island and continental mountains. Or am I misunderstanding this statement?

- Related, on lines 161-161, authors state that: 'The strength and continuity in the diurnal temperature range-species' range size relationship in island and continental mountains (Fig. 3A) suggest that no additional island-specific factors (e.g. lower competition) are needed to explain the range-size pattern'. This I find a very strong statement, as it seems to imply that this one relationship explains all of the variance, which is, especially for continental mountains, all but true (the R^2 of that relationship is not even that high). There are definitely other explanatory factors to 1) explain the remaining variance, and 2) explain the already explained variance due to underlying correlations with the parameter of interest. (I might also be misinterpreting this sentence, as it does mention 'island-specific factors' yet seems to be dealing with relationships in island AND continental mountains)

- The thoughts above are reflected in the fact that the patterns don't hold up at the smaller scale. However, this doesn't take away from the fact that the relationship is clearly there and that it is, at the global scale, contradicting the original hypothesis by Stevens, which I find a very important observation, and that the 'temperature range squeeze' hypothesis is a theoretically very appealing one. Nevertheless, it is important that this 'temperature range squeeze' hypothesis doesn't work at finer scales. And, I agree with Reviewer 3 that there are other possible hypotheses at play. For example, we know that lower elevational limits are often not driven by climate but by competition: many alpine species can very well live in warmer environments (see their success in rock gardens in the lowlands, and see also our observations that when competition is reduced, in roadsides, that they are moving downhill

substantially (Lembrechts et al. 2017). This fact suggests that observed elevational limits are only on one side driven by 'lethal temperatures', which is contradictory with the core of the 'temperature range squeeze' hypothesis. This at least needs a significant discussion in the text, why the 'lower elevation limits are competition-driven'-hypothesis is not applicable, especially as islands often have 1) lower competition, and 2) altered disturbance regimes (e.g., volcanoes, hurricanes) that both could result in broader elevational ranges.

- Finally, the issue of disturbance mentioned in Lembrechts et al. 2017 also highlights the importance of small-scale variation: roadside disturbance easily altered elevational range limits with 200 elevational meters (highland natives) up till 600 elevational meters (lowland non-natives). One could assume data in the underlying study is coming from undisturbed sites, but that also might need discussion.

Minor point:

- I wasn't 100% sure what the '0 to 1980 AD' signified, could you clarify in the text?

To conclude, I do think the paper is important; challenging and questioning these kind of ecological theories is critical and relevant to a broad audience. And while the paper might not be able to fully answer to the mechanics behind the 'temperature range squeeze'-hypothesis, and some more nuance throughout the text might be welcomed, I do think that the story holds up as it is written now. Indeed, statements like the title (Diurnal temperature range as a key predictor of plants' elevation ranges globally) do make sense given the global pattern, even though the full mechanism behind it is not disentangled. I think my main issue might be with the fact that very often, only one side of a species elevational niche is considered to be limited by (lethal) temperatures, which as far as I understand doesn't match with the 'temperature range squeeze'-hypothesis proposed here.

Kind regards,

Jonas Lembrechts

References

Lembrechts, J. J., Alexander, J. M., Cavieres, L. A., Haider, S., Lenoir, J., Kueffer, C., ... &

Milbau, A. (2017). Mountain roads shift native and non-native plant species' ranges. *Ecography*, 40(3), 353-364.

Maclean, I. M., & Early, R. (2023). Macroclimate data overestimate range shifts of plants in response to climate change. *Nature Climate Change*, 1-7.

We are grateful to the additional reviewer for bringing his expertise and insight on the manuscript. We detailed below our responses to the comments and how we addressed the criticism.

I got 'called in' as an additional expert to weigh in on the remaining issue in the underlying paper, regarding the question of the climate data at hand is suitable to answer the research question posed here. I will start by saying that I really liked the paper, that it was very easy to read, and very elegantly written. It is always nice to see papers aimed at confirming or disproving general ecological theory, so I applaud the authors for their attempt!

Thank you for your enthusiasm, we are glad you liked the paper.

However, the debate between authors and reviewer is an important one, and I hope I can contribute with some constructive thoughts: - I follow reviewer 3 that CHELSA climate data doesn't give a very good proxy of local conditions as experienced by plants in the mountains, due to its coarse resolution and it only being relevant for free-air conditions and not in-situ temperatures. This is especially relevant actually for diurnal temperature ranges, which are strongly affected by the buffering of 1) topography and 2) vegetation. By using the spatial heterogeneity in diurnal temperature ranges, which can be huge in mountain areas, plants can avoid many of the coarse-resolution 'lethal temperatures'. A good paper in that regard is Maclean and Early 2023, which provides a neat theoretical figure on how species can live in particular fine-grained pixels within a coarse-grained pixel that is in theory unsuitable. Of course, this doesn't mean that overarching macroclimatic patterns cannot be detected – we have been doing macro-ecology with macroclimate data successfully for years – but it is at least an important discussion point to add to the paper.

We agree that resolution of the climate data might not be high enough to capture micro variations in temperature, which can be another reason why the local-scale analyses were less conclusive. To reflect this better we specified on lines 183-185 that the weak influence of temperature variation "could be exacerbated in the local-scale analyses by the resolution of the climate data that might hide spatial heterogeneity in temperature variation along single elevation gradients", with mention of the Maclean paper.

However, micro variations of diurnal temperature range (occurring within single or a few climate

pixels) would be too localised to have any influence on species' range sizes averaged to an entire mountain.

- Related to the above, the actual experienced diurnal pattern would be significantly different for forest understory versus open area species. And as long as the buffering of forests is the same everywhere, your macroclimate relationship would hold up. However, this is not the case, a similar forest cover could be buffering temperatures more in a wet than in a dry climate, for example. However, I don't think this would necessarily affect the observed patterns.

We agree that it could be possible that the buffering influence of forest can vary between wet and dry forests. As pointed out by the reviewer, we do not expect such differences between dry and wet forests to have much influence on the observed pattern.

- To me, something interesting and important is going on in Fig. 2a: the strong pattern in relationship with diurnal temperature range seems largely driven by the island vs mainland dichotomy. Indeed, within mainland sites, the relationship with diurnal temperature range is significantly weaker. This in itself is possible, but the authors state 'diurnal temperature range was the only variable to account for variation in species' range sizes between island and continental mountains.' (L156-157) I think this should be nuanced to 'the only variable that we tested for'.

We presume you mean Fig. 3a. We replaced "only variable" with "only tested variable" on line 157.

As I can imagine, in line also with reviewer 3's thoughts, I believe, that other parameters (for example continentality, to name just one, or simply distance to the ocean) would also account for the variation in species' range sizes between island and continental mountains. Or am I misunderstanding this statement?

This is correct. In fact, the diurnal temperature range gradient in Fig. 2 and 3 is essentially a oceanic-continental climate gradient.

- Related, on lines 161-161, authors state that: 'The strength and continuity in the diurnal temperature range-species' range size relationship in island and continental mountains (Fig. 3A) suggest that no additional island-specific factors (e.g. lower competition) are

needed to explain the range-size pattern'. This I find a very strong statement, as it seems to imply that this one relationship explains all of the variance, which is, especially for continental mountains, all but true (the R^2 of that relationship is not even that high). There are definitely other explanatory factors to 1) explain the remaining variance, and 2) explain the already explained variance due to underlying correlations with the parameter of interest. (I might also be misinterpreting this sentence, as it does mention 'island-specific factors' yet seems to be dealing with relationships in island AND continental mountains)

We agree that other variables might have an influence on the observed pattern. We rephrased that sentence with "suggest that short-term temperature variation plays a major role in driving the observed range-size pattern that cannot be imputed to island-specific factors (e.g. lower competition)" on lines 162-164.

- The thoughts above are reflected in the fact that the patterns don't hold up at the smaller scale. However, this doesn't take away from the fact that the relationship is clearly there and that it is, at the global scale, contradicting the original hypothesis by Stevens, which I find a very important observation, and that the 'temperature range squeeze' hypothesis is a theoretically very appealing one. Nevertheless, it is important that this 'temperature range squeeze' hypothesis doesn't work at finer scales. And, I agree with Reviewer 3 that there are other possible hypotheses at play. For example, we know that lower elevational limits are often not driven by climate but by competition: many alpine species can very well live in warmer environments (see their success in rock gardens in the lowlands, and see also our observations that when competition is reduced, in roadsides, that they are moving downhill substantially (Lembrechts et al. 2017). This fact suggests that observed elevational limits are only on one side driven by 'lethal temperatures', which is contradictory with the core of the 'temperature range squeeze' hypothesis. This at least needs a significant discussion in the text, why the 'lower elevation limits are competition-driven'-hypothesis is not applicable, especially as islands often have 1) lower competition, and 2) altered disturbance regimes (e.g., volcanoes, hurricanes) that both could result in broader elevational ranges.

Yes, as we specified on lines 82-84, the squeeze of species' ranges might not be symmetrical and can be stronger at higher elevations because freezing temperatures are often a stronger limiting factor than warm temperatures in plants. We agree that the lower limit of species' ranges is likely to be codetermined by other factors, including interspecific interactions and/or

human disturbances. We clarified this on lines 82 and 84. It is also likely that other variables have stronger influences at a local scale given the relatively small range of thermal variation within single mountains, as specified on lines 181-183.

The temperature range squeeze hypothesis as presented in Fig. 1 is purely theoretical and shows how temperature variation should logically influence species' range sizes in the absence of other factors. It appears that at a global scale, our data follow that theoretical schema, i.e. the larger the temperature variation, the smaller species' range sizes. What happens in the lower limit doesn't contradict that schema as the hypothesis doesn't state that the pattern is necessarily symmetrical. For instance, interspecific interactions at the lower end of species' ranges is likely to shift the lower boundary of those ranges to higher elevations but the range squeeze would still happen at the higher end of species' ranges where competition is expected to be weaker (unless of course the higher limit is determined by physical barriers such as a mountain top).

In addition, we could also hypothesise a link between competitiveness and temperature variation. E.g. we could imagine that species would become more vulnerable to competitors as they grow further away from their temperature optimum, which would have different implications between high and low diurnal temperature range mountains.

Finally, the fact that the negative trend is visible in both island and continental mountains and that the relationship between both land types is continuous indicates that competition does not drive the pattern, or at least is not necessary to explain it. We clarified this point on lines 161-162.

- Finally, the issue of disturbance mentioned in Lembrechts et al. 2017 also highlights the importance of small-scale variation: roadside disturbance easily altered elevational range limits with 200 elevational meters (highland natives) up till 600 elevational meters (lowland non-natives). One could assume data in the underlying study is coming from undisturbed sites, but that also might need discussion.

We agree that local human disturbances are likely to have a strong influence on species' range sizes. We added the Lembrechts et al. 2017 reference on line 187 replaced "land use" with "human disturbances" on line 186 to support this. However, because of the relatively localised disturbances of roads on species, we believe this is mostly an issue for the local scale analysis.

Minor point: - I wasn't 100

We replaced "0 to 1980 AD ($\Delta\text{MAT}_{0-1980}$)" with "in the last 2000 years (from 0 to 1980 AD; $\Delta\text{MAT}_{0-1980}$)" on line 109.

To conclude, I do think the paper is important; challenging and questioning these kind of ecological theories is critical and relevant to a broad audience. And while the paper might not be able to fully answer to the mechanics behind the 'temperature range squeeze'-hypothesis, and some more nuance throughout the text might be welcomed, I do think that the story holds up as it is written now. Indeed, statements like the title (Diurnal temperature range as a key predictor of plants' elevation ranges globally) do make sense given the global pattern, even though the full mechanism behind it is not disentangled. I think my main issue might be with the fact that very often, only one side of a species elevational niche is considered to be limited by (lethal) temperatures, which as far as I understand doesn't match with the 'temperature range squeeze'-hypothesis proposed here.

Thank you for your criticism, we hope the revised version nuances the text appropriately.